# Comparative Effectiveness of Filamentous Fungi in Biocontrol of *Meloidogyne javanica* and Activated Defense Mechanisms on Tomato

**DOI:** 10.3390/jof9010037

**Published:** 2022-12-26

**Authors:** Lobna Hajji-Hedfi, Wassila Hlaoua, Awatif A. Al-Judaibi, Abdelhak Rhouma, Najet Horrigue-Raouani, Ahmed M. Abdel-Azeem

**Affiliations:** 1Regional Centre for Agricultural Research of Sidi Bouzid, CRRA, B.P. 357 Gafsa Road Km 6 Sidi Bouzid, Sidi Bouzid 9100, Tunisia; 2Higher Agronomic Institute of Chott-Mariem 4042, Sousse University, Sousse 4000, Tunisia; 3Department of Biological Sciences-Microbiology Section, Faculty of Science, Jeddah University, Jeddah 21959, Saudi Arabia; 4Department of Botany and Microbiology, Faculty of Science, University of Suez Canal, Ismailia 41522, Egypt

**Keywords:** root-knot nematode, BCAs, SCUF, nematicide, defense responses, enzymatic activities

## Abstract

The nematicidal potential of five filamentous fungi as biological control agents (BCAs) against the root-knot nematode (RKN), *Meloidogyne javanica*, infecting tomato was assessed in vitro and in pot experiments. The five promising native taxa, namely *Trichoderma longibrachiatum*, *T. harzainum*, *T. asperellum*, *Lecanicillium* spp., and *Metacordyceps chlamydosporia*, were selected to compare their effectiveness against both chemical (Mocap, 10% ethoprophos) and biological (abamectin) nematicides on *M*. *javanica* reproduction indices and plant growth parameters. The stimulation of defense mechanisms was assessed by monitoring changes in the enzymatic activities of the polyphenol oxidase (PPO), peroxidase (POD), ascorbate peroxidase (APX), catalase (CAT), lipid peroxidation (MDA), phenols, and proteins content of tomato roots. The laboratory assays revealed that *T. longibrachiatum, M. chlamydoporia*, and *Lecanicillium* spp. seemed to be the most effective under laboratory conditions, with more than 60% of juvenile mortality. The egg infection rate was above 62%, and the egg hatching rate was below 32%. The direct parasitism by the five taxa was confirmed by scanning electron microscope observation. The results of this study found a similar parasitism mechanism for *T. longibrachiatum, T. harzianum, and M. chlamydosporia,* where their hyphae and spores adhered to the *M. javanica* juveniles cuticle layer and formed trapping rings around them. The pot experiment results showed that *T. harzianum* and *Lecanicillium* spp. enhanced the plant growth parameters. *Trichoderma longibrachiatum*, abamectin, and the ethoprophos-based nematicides effectively decreased the reproduction rates of the nematode. The *Trichoderma* species and *M. chlamydosporia* significantly reduced the gall index and female fecundity of RKN. The treatment with BCAs and chemical nematicides involved a significant increase in the antioxidant activities of nematode-infected plants. The ethoprophos and fungal treatments decreased the MDA and total phenols content compared with the nematode-infested seedlings. This paper analyzes the advancements made towards the effective and efficient biocontrol of *M. javanica* using different fungal taxa, especially *T. longibrachiatum* and *M. chlamydosporia,* and the implications of these advancements for sustainable agriculture and food security.

## 1. Introduction

Root-knot nematodes (RKNs) of the *Meloidogyne* spp. are plant parasites that infect more than 2000 plant species worldwide and belong to the order Tylechnida and family Meloidogynidae [1]. These organisms are economically destructive, causing significant damage to cultivated crops such as potatoes and tomatoes [2,3]. In Tunisia, RKNs are responsible for causing up to 60% yield losses in tomatoes under greenhouse conditions [4], so effective methods for managing them are a necessity [5,6]. Overall, the biological control of such plant pests and diseases has attracted increasing attention amid efforts to find alternatives to chemical pesticides [7,8,9]. Improving the antagonistic soil potential is proving to be a cost-effective, eco-friendly, target-specific, and sustainable approach to the management of plant parasitic nematodes [10,11].

The beneficial properties of soil microbiota for biocontrol are well-known since they reside in the rhizosphere of host plants [12], where they play a key role in plant development and responses to various biotic and abiotic stresses [13]. Currently, producers rely on these microorganisms to improve plant health in the context of sustainable production systems [14,15]. Among the fungi that thrive in the rhizosphere and have shown biocontrol potential are species of the genera *Paecilomyces*, *Trichoderma*, and *Verticillium* [16,17,18]. *Paecilomyces* is a cosmopolitan fungus known mainly for its nematophagous capacity that has been identified as a bio-stimulant of plant growth and crop yields [19]. *Trichoderma* spp. are ubiquitous in nature and spread rapidly, and they account for nearly 70% of the fungal biocontrol agents (BCAs) market [7]. In particular, growers use *Trichoderma harzianum* extensively to control several soil- and seed-borne phytopathogens and as a biofertilizer [7,9,20]. Among *Verticillium* spp., biocontrol activity against several pests and diseases has been documented, leading to their commercial use [21]. Thus, *Pochonia chlamydosporia* (=*Verticillium chlamydosporium*), a facultative parasite of the eggs of sedentary cyst nematodes and RKNs, has been widely studied as a microbial control agent [22,23,24].

Based on the aforementioned information, successful biocontrol requires careful selection of microbial agents because of the intraspecific variation [25] and the interactions of BCAs with soil biota and ecology [26]. The soil microorganisms employ direct and indirect antagonism mechanisms, including competition for space and nutrients, mycoparasitism, antibiosis, plant growth promotion, and induction of plants’ defense responses [10,12,24,27,28,29]. The defense responses developed by plants against stress are complex and involve numerous physiological, molecular, and cellular adaptations.

Among the biotic constraints hampering tomato production are the RKNs of the *Meloidogyne* spp. These organisms spend most of their life cycles embedded in the roots of their hosts as plant endo-parasitic nematodes and are, therefore, exposed to a variety of host defense responses, including the generation of damaging reactive oxygen species (ROS) [30,31]. ROS are produced naturally in the living cell and at an increased rate during cell cycles. They modify cellular components and, at high concentrations, cause oxidative damage to proteins, DNA, and lipids [32,33]. However, a growing body of evidence indicates that plants also make use of ROS as signaling molecules for regulating various physiological and biological processes relating to growth, death, development, and reactions to biotic and environmental challenges [34,35,36,37]. In order to limit the harmful oxidant effects of ROS, organisms’ enzymatic and non-enzymatic systems act as defense mechanisms to maintain efficient cell functioning [38,39]. Antioxidant enzymes represent a major ROS-scavenging force in plant cells, and the changes in these enzymes are crucial for tolerance to both biotic and environmental stress [39,40,41].

The objective of this study was to validate and compare the effectiveness of native isolates of various nematophagous fungi against *M. javanica*. The analysis served to evaluate certain plant defense barriers consequently to the presence of these fungi in comparison with some commercially available chemical and biological nematicides in Tunisia.

## 2. Materials and Methods

### 2.1. Sampling

Samples of infected tomato plants cultivated in Chott Meriem, Tunisia (35°56′17″ N, 10°33′18″ E) were collected and transferred to the laboratory for further investigation. Infected roots were used for isolation of the RKN (*M. javanica*) and maintained on the susceptible tomato cultivar (Rio Grande) to produce enough inoculum for further experiments. Soil samples were collected in sterile plastic bags from RKN-infested fields in Monastir (35°76′43″ N, 10°8′11.3″ E) and Kairouan (35°67′12″ N, 10°10′05″ E) regions, Tunisia, respectively. The sampled soils were selected for low infestation of nematode for potential antagonistic microorganisms contributing to nematode suppression. Then, they were transferred directly to the laboratory and kept in the refrigerator until plating out.

### 2.2. Nematode Inoculum

Identification of the RKN, *Meloidogyne javanica* (Treub) Chitwood, based on morphological characteristics of the perineal pattern of the adult females according to Taylor and Sasser [42]. The egg masses were collected individually from the roots, gently washed to remove the soil debris, and macerated in 0.5% (*v*/*v*) NaClO, as described by Hussey and Barker [43]. Eggs were placed in Baermann trays [44] to obtain second-stage juveniles (J2). A counting chamber under light microscopy (Leica DM300-GmbH, Wetzlar, Germany) served to determine the J2 number used for the experiment.

### 2.3. Isolation, Phenotypic and Molecular Identification of Fungi

For the isolation of filamentous fungi, the dilution soil plate technique [45] was used. Czapek’s yeast extract agar (30 g/L sucrose; 3.0 g/L sodium nitrate; 0.5 g/L potassium chloride; 0.5 g/L magnesium sulfate heptahydrate; 0.01 g/L iron(II) sulfate heptahydrate; 1.0 g/L di-potassium hydrogen phosphate; 5.0 g/L yeast extract; 15.0 g/L agar agar) and potato dextrose agar (PDA) were used as isolation media. Both media were supplemented with Rose Bengal (1/15,000) and chloramphenicol (50 ppm) for the suppression of bacterial growth [46]. The plates were incubated (BJPX-HTBII, Biobase, Jinan, China) at 25 ± 2 °C for 7 days, and then the developing colonies were identified.

Taxonomic identification of isolated fungi used the phenotypic approach down to the species level on standard media based on the following identification keys: *Pochonia* [47,48], *Trichoderma* [49], for *Verticillium* [50]. The microscopic characteristics were observed with a Carl Zeiss amplival microscope (GmbH, Wetzlar, Germany). The systematic arrangement in the present list follows the latest system of classification appearing in the 10th edition of Ainsworth and Bisby’s Dictionary of the Fungi [51]. Name corrections, authorities, and taxonomic assignments of all taxa reported in this work were checked against the Index Fungorum database (www.indexfungorum.org, accessed on 5 December 2022). For molecular characterization, Fungal taxa were maintained on potato dextrose agar (PDA) containing 150 mg/L streptomycin and placed in an incubator for 10 days at 26 ± 2 °C. Genomic DNAs were extracted from 7–10 day-old cultures as described by White et al. and using the universal primers ITS4/ITS1 [52]. PCR amplifications were performed according to the procedure of Glass and Donaldson [53]. The amplification of the products was analyzed by electrophoresis using a 1.5% agarose gel (Sigma Aldrich, Saint Louis, MO, USA) stained with SYBR Safe DNA Gel Stain (Invitrogen, Carlsbad, CA, USA). Then, PCR-purified products were sequenced (Applied Biosystems, Bedford, MA, USA). The obtained sequences were compared with reference sequences of GenBank databases using the Blastn tool of the National Center for Biotechnology Information (NCBI). Identification was considered when homology percentages were greater than 98% between the submitted sequences and the reference ones and registered with accession number in the NCBI Database.

### 2.4. In Vitro Evaluation of Direct Fungal Parasitism of M. javanica

In vitro tests were carried out to determine the direct effect of the studied fungi on *M. javanica* egg hatching and parasitism, and second-stage juveniles’ mortality. The five fungal isolates were previously grown on PDA for 6–10 days at 28 °C, and a 5 mm disc of each fungus was transferred into 2% water agar plates containing 1% ampicillin (100 μg mL^–1^). Then, 100 surface-sterilized *M. javanica* eggs and freshly hatched juveniles were added for ovicidal and larvicidal assays. The plates were incubated at 26 ± 2 °C for 7 days. Control treatment consisted of plates without growing fungi. After 7 days of incubation, eggs and juveniles were observed under a stereomicroscope (Leica M80, Schweiz AG, LED 2500, Taipei city, Taiwan). The numbers of hatched juveniles and infected eggs (colonized with fungi hyphae) were counted. The hatching and infection rates of *M. javanica* eggs were recorded. Regarding the juveniles test, the numbers of mobile and immobile J2s were counted under a stereomicroscope, and juvenile mortality rates were determined. The egg and juvenile in vitro tests were performed in triplicate with 6 replicates for each treatment.

### 2.5. Scanning Electron Microscopy (SEM)

The fungal-infected *M. javanica* eggs and juveniles of the in vitro experiment were prepared for SEM as described in the Karnovsky’ protocol [54]. Observations were made, and photographs were taken using a Hitachi SU3500 scanning electron microscope (Toranomon, Minato-ku, Tokyo 105-6409, Japan).

### 2.6. Pot Experiment

Pots with a 12 cm diameter (1 L) were filled with 1 kg autoclaved soil mixture (loamy soil, sand, and peat 1:1:1; *v*/*v*). Tomato cv. Rio Grande seeds were grown for 15 days in a pot tray, and four-leaved seedlings were transferred to the pots for inoculation. Two days after transplantation, the seedlings were inoculated with a 1-mL suspension containing 1500 J2 and poured into two holes around each plant root in the pot.

Immediately after the nematode inoculations, the tomato plants were treated with either the BCAs or chemical nematicides. The healthy seedlings that served as controls received neither the nematode inoculations nor the BCA or chemical nematicide treatments. The treatment groups included (i) plants inoculated with *M. javanica* (RKN), (ii) plants inoculated with nematodes and treated with all recovered taxa during the study, (iii) plants inoculated with nematodes and treated with a chemical nematicide based on ethoprohos (CHM), and (iv) plants inoculated with nematodes and treated with a biological nematicide based on the abamectin produced by *Streptomyces avermitilis* (BIO). Mocap (10% ethoprophos; BAYER Cropscience, Rhone-Poulenc, Inc., Strasbourg, France) was added to soil 2 days after transplantation of the seedlings, at 0.5 g per pot, corresponding to the recommended rate of 50 kg/ha. Tervigo (020SC, abamectin 20 g/L, Syngenta Crop Protection AG), a suspension concentrate of abamectin, was applied at 0.166 mL per pot by pipette on the soil surface, corresponding to the recommended dose of 5 L/ha. For BCAs fungi, the species were grown on PDA at 25 ± 2 °C for five days until sporulation, and then, an Erlenmeyer flask containing 50 mL of potato dextrose broth was inoculated with four pieces individually. Spore production was induced in an orbital shaker, and the spores were recovered from culture by filtration. A hemocytometer served to adjust the concentrations of the spores to 4 × 10^6^ spores/mL. Ten milliliters of the spore suspensions were poured into two holes around the rhizospheres of the tomato plants. The pots were arranged in a randomized complete block design with 10 replicates per treatment, and the entire experiment was repeated three times. The tomato seedlings were maintained in a greenhouse at 25 ± 3 °C with 65 ± 5% relative humidity and periodically watered and fertilized with a nutrition solution [55]. Three replicates of plants per treatment were uprooted 7 days after the nematode inoculation and were washed with distilled water to carry out the biochemical analysis. The seven remaining plant replicas were removed 60 days after inoculation to measure the growth parameters in terms of plant and root length and weight. The height of the plants was measured from the stem base (above ground) to the tip of the longest leaf or panicle. The root length was measured from the stem base (below ground) to the root system cap.

#### 2.6.1. Antioxidant Enzyme Activities in Tomato Roots

One gram of root material of each plant sampled 7 days after the RKN inoculations was ground with 5 mL cold 50 mM phosphate buffer (pH 7.0), PMSF 1 mM, EDTA 0.2 mM, 1% PVP, and 0.1% Triton X-100 in a pre-chilled mortar and pestle. The homogenate was centrifuged at 12,000× *g* at 4 °C for 10 min [13]. The supernatant was stored at −80 °C (Biobase Ultra-Low Temperature Freezer, Jinan, China) and used to determine the enzymatic activities.

##### Peroxidase Activity (POD)

Peroxidase activity was evaluated based on Δ absorbance following the oxidation of pyrogallol in the presence of H_2_O_2_ according to the modified method of Reddy et al. [30]. The assay mixture for peroxidase contained 1.5 mL of 0.05 M pyrogallol, 0.5 mL of 1% H_2_O_2_, and 0.5 mL of enzyme extract. Spectrophotometric readings were taken using a UV2005 spectrophotometer (JP Selecta, Barcelona, Spain) at a wavelength of 420 nm for 3 min at 30 s intervals. One unit of peroxidase activity is defined as the changes in absorbance of the reaction mixture per minute (unit mn^−1^ g^−1^ of FW).

##### Catalase Activity (CAT)

Catalase activity was assayed by adding 0.05 mL of enzyme extract to 1.5 mL of phosphate buffer (pH 7.0), 0.95 mL of distilled water, and 0.5 mL of H_2_O_2_. The decomposition of H_2_O_2_ was monitored for 60 s at 240 nm in a spectrophotometer at a temperature of 25 °C [56].

##### Ascorbate Peroxidase (APX)

Ascorbate peroxidase activity was estimated according to the modified method of Nakano and Asada [57]. The reaction medium consisted of 890 µL of 50 mM phosphate buffer (pH 7.0), 2 µL of 0.1 M ascorbate, 30 µL of enzyme extract, and 20 µL of 10 mM H_2_O_2_. APX activity was determined by monitoring the rate of oxidation of ascorbate at 290 nm in a spectrophotometer at 30 °C for 3 min.

##### Polyphenol Oxidase (PPO)

Polyphenol oxidase activity was determined according to the modified method of Mayer et al. [58]. It was estimated by adding 1.5 mL of 100 mM sodium phosphate buffer (pH 6.5), 200 µL enzymes extract, and 200 µL of 0.01 M catechol. The changes in the optical density were observed at 450 nm every 30 s for 3 min.

#### 2.6.2. Total Soluble Protein Content

The root total soluble protein was extracted by the TCA/acetone method as described by Xu et al. [59]. The total protein content was determined according to the method by Bradford [60]. Briefly, 5 µL of root extract supernatant was added to 795 µL phosphate buffer and 200 µL Bradford reagent. Absorbance was measured at 595 nm using a spectrophotometer. The total protein content in the tomato roots was determined with reference to the standard curve produced by known concentrations of bovine serum albumin (BSA).

#### 2.6.3. Total Phenol Content

One gram of root samples was homogenized in 2 mL 80% methanol with a mortar and pestle, and the homogenate was centrifuged (BKC-TH12R, Biobase, Jinan, Shandong, China) at 10,000× *g* for 30 min, after which the supernatant was collected. Then, 100 µL of methanolic extract was added to 500 µL Folin–Ciocalteau reagent and 2 mL of Na_2_CO_3_ solution. The mixture was incubated for 3 min at room temperature, followed by heating in a water bath at 70 °C for 20 min in darkness [61]. Spectrophotometric readings were taken at 760 nm. The final concentrations of phenols were expressed in µg/g fresh root weight derived from a standard curve obtained with gallic acid (0.01, 0.05, 0.1, 0.2, and 0.3 mg/mL).

#### 2.6.4. Lipid Peroxidation (MDA Content)

The determination of the lipid peroxidation was conducted by homogenizing 0.2 g of roots in 2 mL trichloroacetic acid, thiobarbituric acid, and HCl, followed by centrifugation at 12,000× *g* for 20 min. The reaction product was then heated at 95 °C for 30 min, followed by cooling on ice. The reaction product was centrifuged at 12,000× *g* for 15 min, and the absorbance of the supernatant was measured at 532 and 600 nm [62]. The MDA equivalent was derived from the absorbance, and the concentration of MDA was determined by its molar extinction coefficient of 155 mM^−1^ cm^−1^; the results were expressed as μmol/g f.w.

#### 2.6.5. Extraction and Enumeration of Nematodes and Assessment of Their Development

Root galling was rated according to the scale proposed by Hussey and Janssen [63], with a value of 0 indicating no galling, 1 indicating trace infection with a few small galls, 2 indicating ≤ 25% of the roots galled, 3 indicating 26–50% galled, 4 indicating 51–75% galled, and 5 indicating > 75% galled. The gall number per g of the root was calculated using a homogenized subsample of 5 g of roots from each plant. The total root nematode population was determined by the blender centrifuge method [64]. In addition, the total soil nematode population was determined using the modified flotation–centrifugation technique [65]. The reproduction rate of the nematodes (Pf/Pi) was then calculated by dividing Pf (the sum of total root and soil nematode populations) by the number of nematode juveniles inoculated (Pi: 1500 J2/plant). At the end of the experiment, the egg masses were collected and incubated at 27 °C for the egg-hatching test. Two egg masses were laid on a micro-sieve (with 40 µm pores), placed on 5 cm diameter Petri plates, and submerged in distilled water. The Petri plates were maintained at 25 ± 3 °C in darkness. The J2 number was counted every 72 h for seven days. Each treatment was repeated three times, and the test was carried out twice [66].

### 2.7. Statistical Analysis

All of the data were subjected to a one-way analysis of variance (ANOVA) and checked for homogeneity of variance before being pooled, and the means were compared using Tukey’s multiple-range test. Differences at *p* ≤ 0.05 were considered significant. The statistical analysis was performed using SPSS version 20 software for Windows.

## 3. Results

### 3.1. Isolation and Identification of Filamentous Fungi

Five taxa were isolated and phenotypic identified during this study, namely: *Trichoderma asperellum* Samuels, Lieckf., and Nirenberg; *T. harzianum* Rifai; *T. longibrachiatum* Rifai; *Metacordyceps chlamydosporia* (Evans) Sung, Sung, Hywel-Jones, and Spatafora; and *Lecanicillium* spp. Recovered taxa were assigned to a single class (Sordariomycetes), single order (Hypocreales), and three families (Hypocreaceae, Clavicipitaceae, Cordycipitaceae), respectively.

All recovered taxa were preserved in the Suez Canal University Fungarium (SCUF) under deposition numbers from SCUF0000346-SCUF0000350 (https://ccinfo.wdcm.org/details?regnum=1180, accessed on 2 September 2022).

The analysis of obtained ITS rDNA sequences revealed their close homologies with *Trichoderma harzianum* (MK007293, 99%), *T. longibrachiatum* (LT707585, 98%), *T. asperellum* (MN396592, 99%), *Metacordyceps chlamydosporia* (AB214654, 99%), and *Lecanicillium* spp. (MK732148, 98%). The sequences were submitted to Gen-Bank and assigned under the accession numbers: OP799678, OP799680, OP799679, OP799677, and OM169327, respectively.

### 3.2. In Vitro Experiments

Data presented in Table 1 indicated clearly that the five treatments with fungal isolates exerted highly significant nematicidal activity (<0.01). Compared to the control, all examined fungi increased juvenile mortality and egg infection rate and decreased the egg-hatching rate (Table 1). Obtained results of juvenile mortality indicated that *T. longibrachiatum* seemed to be the most effective treatment (87.43%). Both *T. harzianum* and *M. chlamydosporia* were also effective, with 60.27 and 62.28%, respectively (Table 1).

The egg-hatching rate reveals the potential for population growth of *M. javanica* after treatment with different fungal isolates, which is strongly combined with future damage to the nematode. The lowest egg-hatching rate of *M. javanica* was seen in the *Lecanicillium* spp. treatment (30.05%), which was less than one-third of what was observed in the control (95.69%). A 32.40% hatching rate was observed in the *M. chlamydoporia* treatment (Table 1).

All examined fungi were able to infect more than 50% of *M. javanica* eggs. The highest egg infection rate was registered for *M. chlamydoporia* (86.84%), followed by *Lecanicillium* spp. (81.36%). However, the lowest infection was recorded for *T. asperellum* (66.29%) (Table 1).

### 3.3. Parasitism Observation

The SEM observation of RKN larvae showed that *T. harzianum, T. longibrachiatum,* and *M. chlamydosporia* coiled on and around the juveniles (Figure 1A,D). The three fungal species, hyphae, and conidia, adhered to the juveniles’ cuticle layer. They formed trapping rings around RKN juveniles, and they caused their immobility. The fungal spores of *T. longibrachiatum* and hyphae *T. harzianum* grew into juvenile cuticles (Figure 1B,C). The electron micrographs confirmed that *T. longibrachiatum*, *T. harzianum,* and *M. chlamydosporia* had nematicidal effects, and their pathogenicity mechanism could be illustrated in three successive actions: (i) trapping RKN nematodes by hyphal adhesion into juveniles, (ii) penetration of the cuticle layer, and (iii) presumably digestion of its cellular contents.

### 3.4. Plant Growth Parameters

In general, plant growth improved after treatment with biological or chemical nematicides and after fungal inoculation (Table 2). Depending on the plant growth parameters and compared with the RKN treatment, the various treatments significantly increased the fresh weight of the plants. *Lecanicillium* spp. yielded the longest shoot weight, and *T. harzianum* had the greatest plant height. Only some of the other fungi and chemicals improved root growth. The three *Trichoderma* species significantly increased the root length, while the biological amendment and chemical nematicide yielded the best results for root weight.

### 3.5. Antioxidant Enzymes Activity

Seven days after inoculation with the sedentary nematodes, it was evident that *M. javanica* significantly increased the activity of ROS-scavenging enzymes, especially APX and POD, compared with the control plants. Furthermore, abamectin, the ethoprophos-based nematicide, and the applications of soil fungi amplified the activity of antioxidant enzymes APX, CAT, PPO, and POD, with the rates varying by treatment. The POD maximum activity was recorded in plants inoculated with *T. asperellum* while the other ROS-scavenging enzymes (APX, CAT) increased significantly following soil treatment by *T. longibrachiatum*. Moreover, an increase in the activity of PPO, a resistance-related enzyme, was recorded in the plants that received the chemical treatment ethoprophos (Table 3).

### 3.6. Phenols, MDA, and Protein Content in Tomato Roots after Treatment

The RKNs induced tomato defenses, with a considerable increase in phenols, MDA, and protein content in the roots compared with the untreated control. Moreover, all of the treatments tested significantly decreased these plant defense parameters compared with the RKN alone treatment (Table 4). The MDA content in the tomato roots decreased significantly after the application of all of the treatments except with *Lecanicillium* spp. The *T. asperellum* was associated with the least MDA content in the roots, with 6.97 µmol g^−1^ FW, as well as the lowest quantity of phenols in the plant roots. Likewise, the fungal and nematicide treatments decreased the effect of the RKNs on the protein content, with the plants that received the *M. chlamydosporia* and *T. harzianum* treatments showing the least content (2.50 and 2.54 mg g^−1^ FW, respectively).

### 3.7. Nematode Reproduction

The development of the nematodes was measured as the gall index, the number of galls per g of the root, and the multiplication rate. All of these values decreased significantly more after the tested treatments than the nematode control. The RKN multiplication rate decreased significantly following the treatments with *T. longibrachiatum*, ethoprophos, and abamectin (by 88.5%, 88.3%, and 86.1%, respectively; Table 5). However, the soil and root sterilized surface for re-isolation of the fungal strains confirmed the presence of all of the fungal species tested.

The test for the hatching of the egg masses at the end of the pot experiment revealed the effect of treatments on female fecundity. Figure 2 illustrates the cumulative numbers of J2 obtained after nematode egg hatchings over seven days. The hatching rate of the control trial eggs was higher than those that received the fungal-associated, chemical, and biological treatments. The lowest hatching rate was obtained with *T. longibrachiatum*, followed by ethoprophos, *T. asperellum*, *M. chlamydosporia*, abamectin, *Lecanicillium* spp., and *T. harzianum*.

## 4. Discussion

The results presented in this work indicate that *M. javanica* negatively affected the growth of tomato plants, causing significant reductions in shoot weight and root length. These results are consistent with the effect of *Meloidogyne* spp. on crops such as carrots [67], cassava [68], cucumber [69,70], tomato [71], and potato [72].

Even though synthetic nematicide, such as Mocap or Vydate, may be highly successful against RKNs, using native microorganisms could be seen as a safer and more environmentally friendly alternative. Indeed, the current study illustrated how might the filamentous fungi (BCAs) *T. longibrachiatum*, *T. harzainum*, *T. asperellum*, *Lecanicillium* spp., and *M. chlamydosporia* improved tomato plants growth when they are under *M. javanica* infestation. The marked decrease in gall index, gall number, and nematode multiplication rate caused by the applied strains of fungi (Table 5) and the noticeable reduction in fecundity (Figure 2) prove that they are extremely efficient against RKN.

Among the tested BCAs, *T. harzianum* was the most effective stimulator of tomato growth, followed by the other *Trichoderma* species and *Lecanicillium* spp. *Trichoderma* spp. are generally known to improve plant performance in association with other mechanisms of BCAs, including mycoparasitism, inducing systemic resistance, and producing lytic enzymes [9,73,74].

Among the fungi tested in the present study, *M. chlamydosporia* and *T. longibrachiatum* had the strongest nematicidal effects. Hence, the fungal treatment decreased the multiplication of *M. javanica* on tomato considerably and affected the fecundity of female nematodes. *P. chlamydosporia* (*M. chlamydosporia*) [23,75,76] and *Trichoderma* spp. [77,78,79] have been described as showing promise as bio-control agents against RKN infections in various crops.

The marked decreases in the egg masses and eggs in the root systems caused by the applied beneficial microorganisms are especially significant because this estimate of nematode reproduction, rather than the galling index, is necessary to determine the fecundity of nematodes on plant cultivars. Thus, the overall goal of such BCAs is the identification and deployment of strains that are highly effective against various plant pathogenic fungi and/or nematode pests and can then be developed into registered plant protection products for sale to growers [26].

In biochemical and enzymatic studies of the protein, phenol content, lipid peroxidation, and activity of antioxidative enzymes, such as catalase, peroxidase, ascorbate peroxidase, and polyphenol oxidase, were investigated. The results of the aforementioned studies show a considerable rise in all enzymatic parameters with the exception of MDA, protein, and total phenol content due to the application of different BCAs if compared with *M. javanica* infestation effect on these parameters.

Interactions between plants and parasitic nematodes occur within the context of a “vast molecular plant immunity network” [80]. Following initial contact with host plants, nematodes activate basal immune responses and trigger an oxidative burst in the root tissues [81] that leads rapidly to the production of reactive oxygen species (ROS) and reactive nitrogen species (RNS) as well as toxic molecules derived from secondary metabolism [82]. Many others confirm that one of the first cellular responses following pathogen recognition is the oxidative burst involving the production of ROS [38,83,84,85]. At the same time, the regulation of the antioxidant pathway enables the expression of the genes that encode ROS-scavenging enzymes [39,86,87] in order to prevent intracellular oxidative damage to the host while causing oxidative stress to the pathogen within the host apoplast [88,89]. This regulation explains the finding that *M. javanica* infection was associated with increases in the activity of the ROS-scavenging enzymes i.e., APX, POD, CAT, and PPO related to the defense response mounted by the plant host, which is consistent with the findings of Nagesh et al. [90] and Hajji et al. [30]. Ezzat et al. [91] indicated that plant health is measured by the physiological indices CAT, APX, and PPO. It is widely known that the plant’s tolerance to biotic stress considerably increased once antioxidant enzymes were activated [92]. Moreover, in the context of biotic stress, Ashry and Mohamed [89] observed a significant increase in the activities of POD and CAT in the leaves of flax lines infected with powdery mildew fungi. Likewise, enhanced activities of POD, CAT, and APX were observed in *Vicia faba* leaves infected with bean yellow mosaic virus, indicating that the ROS-scavenging systems may play an important role in managing the ROS generated in response to pathogens [82]. In the same context, Tikoria et al. [93] demonstrated that protein content and enzyme activities of CAT, POD, APX, and PPO were shown to be reduced after nematode infestation but enhanced by vermicompost amendment, which played the same role as our BCAs treatments.

The results also indicated that the nematode infections were accompanied by increases in phenols, lipid peroxidation (MDA), and protein content in the root cells. The infection of various hosts with *M. incognita* [94,95] and other plant–parasitic nematode species (*Rotylenchulus reniformis* and *Tylenchulus semipenetrans*) also resulted in an increase in antioxidant defensive enzymes, phenols, MDA, and protein content [95]. Another study demonstrated that plants infected with the pine wood nematode (*Bursaphelenchus xylophilus*) showed a considerable rise in MDA content [96]. Several researchers have reported that these molecules are directly related to the levels of ROS in plant tissue [83,88,97].

Phenolics are diverse secondary metabolites (including flavonoids, tannins, hydroxycinnamate esters, and lignin) in plant tissues [95] with potent antioxidant properties [83] and can directly scavenge molecular species of active oxygen [98]. Moreover, Michalak [99] presented some evidence of the induction of phenolic metabolism in plants in response to multiple stresses. A recent study on RKN infesting *Oryza sativa* found greater concentrations of total phenols in plants that are identified as phenol-induced lignification of plant epidermis that would protect the plants against nematode invasion [100].

MDA is another of the final products of the peroxidation of unsaturated fatty acids in phospholipids and is responsible for cell membrane damage [100]. Increases in the peroxidation (degradation) of lipids have been reported in plants growing under environmental stresses [101,102,103]. These increases parallel the increased production of ROS, which exacerbate oxidative stress and damage proteins and DNA [104].

The mounting of defenses requires the regulation of a wide variety of proteins at various levels, from the receptors that perceive pathogens through signaling mechanisms to the strengthening of the cell wall or degradation of the pathogen [88]. Plants counteract pathogens by modulating the regulation of the proteins involved in defense or stress responses and ROS metabolism, and protein content decreased by 46% in tomato plants after *M. incognita* infection [105]. The damage of ROS on proteins may include various forms of direct and indirect modification. Direct modification involves the modulation of the activity of proteins through mechanisms such as nitrosylation, carbonylation, the formation of disulfide bonds, and glutathionylation. Indirect modification may involve conjugation with the breakdown products of fatty acid peroxidation [106,107]. Tissues injured by oxidative stress generally contain increased concentrations of carbonylated proteins, which serve widely as a marker of protein oxidation [107].

The results presented here confirm those of Oka et al. [108], who found protein changes at an early stage in the infection of barley and wheat roots with *Heterodera avenae*. The enhanced modification of proteins has been reported in plants under various stresses [87,109,110,111,112].

Another finding of the present study is that treatment of the soil with microorganisms antagonistic to *M. javanica*, in particular, *T. longibrachiatum*, enhanced the activity of ROS-scavenging enzymes compared with the roots inoculated with only the nematodes. The treatment of the soil with the filamentous fungus *T. harzianum*, a well-known BCA, resulted in a significant increase in the specific activities of the resistance-related enzymes peroxidase (POX), polyphenol oxidase (PPO), and phenylalanine ammonia-lyase) compared with plants inoculated with only *M. javanica* [113].

In addition, the treatment of the soil with the BCAs decreased the protein, phenol, and MDA content in the tomato roots seven days after infection with *M. incognita*. The change in the content of MDA (a product of lipid peroxidation) serves as an indicator of the level of oxidative damage that a plant has suffered. The reduction in the MDA content following the treatment with the beneficial microorganisms or another sustainable method (the application of humic acid) has been found to correlate with an increase in antioxidant enzymes [95,114,115]. Several fungi or bacteria act as BCAs, whether directly by reducing nematode development and/or indirectly by stimulating the plant host defense mechanisms involving mainly antioxidant defense enzymes [114,116,117]. Regarding the diverse mechanisms of action, from promoting plant growth and suppressing the reproduction of plant–parasitic nematodes *M. incognita* to inducing resistance, *P. chlamydospria* (*M. chlamydosporia*) and some *Trichoderma* species have been used in an integrated manner to achieve the highest efficiency yet reported [79,118].

## 5. Conclusions

The current investigation demonstrated the positive impact of the treatment of soil with several inborn fungi as BCAs. They presented a strong potential for reducing nematode stress, improving plant growth and development, as well as increasing several bio-active plant antioxidant defense components in tomato plants. This study thus provides insight into the various mechanisms involved in the interactions among plant hosts, nematodes, and antagonistic fungi. The results indicate that, in the future, nematode control using fungal BCAs individually, in consortia, or associated with other sustainable management tools in the context of an integrated approach, can help to control plant–parasitic nematodes in an environmentally and economically beneficial manner.

## Figures and Tables

**Figure 1 jof-09-00037-f001:**
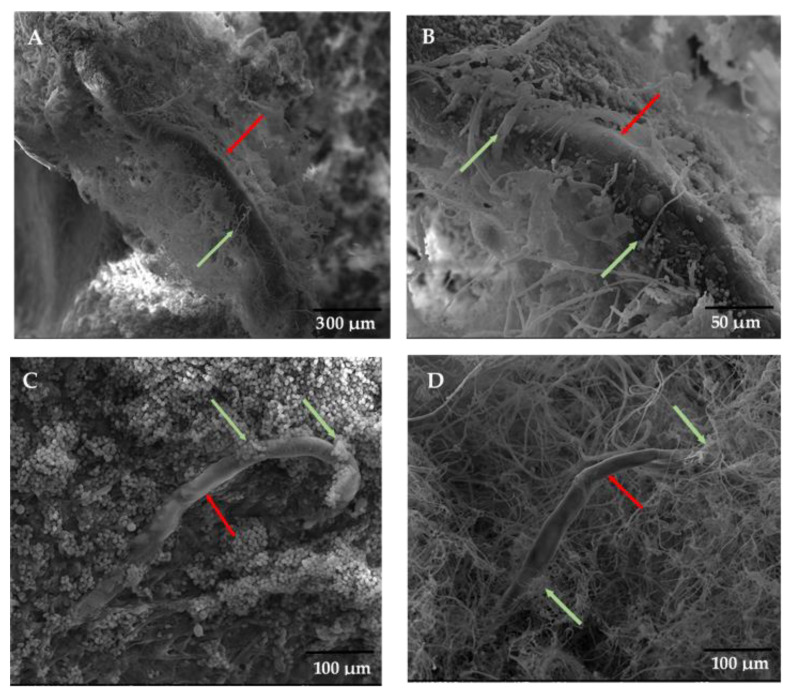
Scanning electron micrographs of *Meloidogyne javanica* second stages juveniles treated with *Trichoderma harzianum* (**A**,**B**), *T. longibrachiatum* (**C**), and *Metacordyceps chlamydosporia* (**D**) ((**A**): scale bars 300 µm; (**B**): scale bars: 50µm; (**C**,**D**): 100 µm; red arrows indicate the nematode juveniles; green arrows indicate the fungus mycelium).

**Figure 2 jof-09-00037-f002:**
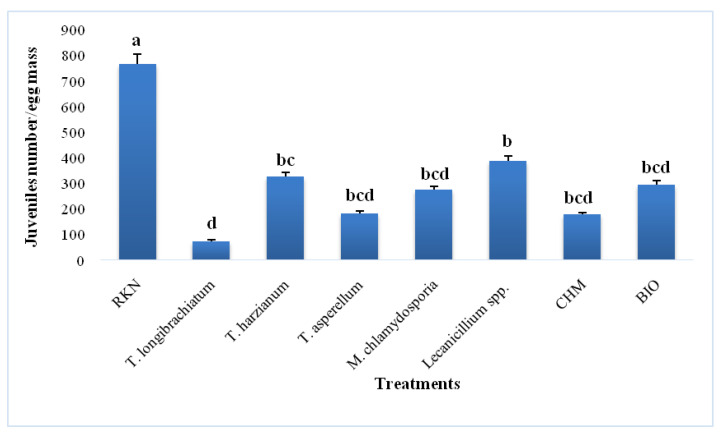
Effect of different treatments on *M. javanica* fecundity at 60 days post-inoculation (Values are mean of 12 replicates (6 egg masses × 2 repetitions) ± SD; RKN: plants inoculated with *M. javanica* only; BIO: plants treated with abamectin; CHM: plants treated with ethoprophos; bars sharing the same letter are not significantly different at *p* ≤ 0.05 to according to Tukey’s multiple range test).

**Table 1 jof-09-00037-t001:** Effects of the nematophagous fungi on *M. javanica* juveniles mortality, egg hatching, and egg infection rate.

Fungal Isolate	Juveniles Mortality (%)	Egg Hatching Rate (%)	Egg Infection Rate (%)
Control	-	95.69 ± 0.58 a	-
*T. longibrachiatum*	87.43 ± 0.70 a	41.20 ± 0.57 d	76.50 ± 0.51 c
*T. harzianum*	60.27 ± 0.76 b	47.94 ± 0.62 c	72.20 ± 1.11 d
*T. asperellum*	44.13 ± 1.58 d	55.21 ± 0.57 b	66.29 ± 0.71 e
*M. chlamydosporia*	62.28 ± 0.48 b	32.40 ± 0.84 e	86.84 ± 0.28 a
*Lecanicillium* spp.	55.84 ± 0.69 c	30.05 ± 0.34 f	81.36 ± 0.33 b

Values are means ±SD (n = 18; 6 replicates × 3 repetitions). A number within the column, followed by the same letter, did not differ statistically. Comparison in the same column (Tukey’s multiple range test at *p* ≤ 0.05).

**Table 2 jof-09-00037-t002:** Effect of soil treatment by filamentous fungi under investigation, biological and chemical nematicides on tomato growth, 60 days after *M. javanica* inoculation.

Treatments	Shoot		Root	
	Weight (g)	Height (cm)	Weight (g)	Length (cm)
Untreated Control	19.80 ± 3.28 ^ab^	36.70 ± 3.35 ^b^	5.02 ± 1.69 ^cd^	20.30 ± 5.50 ^a^
RKN only	11.62 ± 1.24 ^d^	16.92 ± 7.95 ^c^	5.95 ± 0.96 ^c^	18.67 ± 2.34 ^b^
*T. longibrachiatum* + RKN	18.27 ± 2.70 ^b^	36.33 ± 3.71 ^b^	4.22 ± 0.97 ^d^	19.83 ± 4.37 ^a^
*T. harzianum* + RKN	18.65 ± 2.17 ^b^	40.00 ± 6.63 ^a^	6.16 ± 0.58 ^c^	22.50 ± 2.19 ^a^
*T. asperellum* + RKN	19.78 ± 2.50 ^ab^	32.08 ± 4.10 ^b^	4.32 ± 0.29 ^d^	19.83 ± 4.32 ^a^
*M. chlamydosporia* + RKN	15.57 ± 2.92 ^c^	30.50 ± 2.76 ^b^	6.98 ± 1.82 ^b^	18.83 ± 1.33 ^ab^
*Lecanicillium* spp. + RKN	21.32 ± 2.22 ^a^	35.25 ± 4.24 ^b^	7.00 ± 1.84 ^b^	18.08 ± 3.60 ^b^
CHM + RKN	17.00 ± 1.84 ^c^	31.33 ± 6.15 ^b^	7.72 ± 2.59 ^b^	18.50 ± 2.24 ^b^
BIO + RKN	16.03 ± 2.38 ^c^	28.00 ± 3.02 ^b^	9.05 ± 2.07 ^a^	18.08 ± 2.01 ^c^

Values are mean of twenty-one replicates (7 plants × 3 repetitions) ± SD; RKN: plants inoculated with *M. javanica* only; BIO: plants treated with abamectin; CHM: plants treated with ethoprophos; within a column, means sharing the same letter are not significantly different at *p* ≤ 0.05 according to Tukey’s multiple range test.

**Table 3 jof-09-00037-t003:** Activity of antioxidant enzymes: ascorbate peroxidase (APX), peroxidase (POD), polyphenol-oxidase (PPO), and catalase (CAT) in tomato plant roots 7 days after *M. javanica* inoculation.

Treatments	APX	CAT	PPO	POD
	U mn^−1^ g^−1^ FW	U mn^−1^ g^−1^ FW	U mn^−1^ g^−1^ FW	U mn^−1^ g^−1^ FW
Untreated Control	5.91 ± 1.02 ^c^	2.40 ± 0.51 ^d^	1.44 ± 0.37 ^c^	1.57 ± 0.34 ^d^
RKN only	12.02 ± 1.72 ^ab^	5.10 ± 0.80 ^d^	2.64 ± 0.34 ^c^	3.10 ± 0.12 ^c^
*T. longibrachiatum* + RKN	13.22 ± 0.32 ^a^	22.12 ± 0.61 ^a^	6.05 ± 0.41 ^a^	4.13 ± 0.32 ^b^
*T. harzianum* + RKN	11.41 ± 1.01 ^b^	18.64 ± 0.82 ^b^	6.37 ± 0.34 ^a^	3.91 ± 0.22 ^b^
*T. asperellum* + RKN	11.04 ± 0.21 ^b^	19.40 ± 0.34 ^ab^	5.51 ± 0.28 ^ab^	5.92 ± 0.46 ^a^
*M. chlamydosporia* + RKN	13.01 ± 0.20 ^a^	18.96 ± 0.10 ^b^	6.19 ± 0.50 ^a^	4.28 ± 0.40 ^b^
*Lecanicillium* spp. + RKN	10.61 ± 1.02 ^b^	18.82 ± 0.54 ^b^	4.56 ± 0.62 ^b^	3.64 ± 0.21 ^bc^
CHM + RKN	13.11 ± 0.32 ^a^	19.36 ± 0.90 ^ab^	6.31 ± 0.54 ^a^	3.10 ± 0.26 ^c^
BIO + RKN	11.18 ± 0.48 ^b^	10.60 ± 0.83 ^c^	4.32 ± 0.31 ^b^	3.64 ± 0.41 ^bc^

Values are mean of 9 replicates (3 plants × 3 repetitions) ± SD; RKN: plants inoculated with *M. javanica* only; BIO: plants treated with abamectin; CHM: plants treated with ethoprophos; U mn^−1^ g^−1^ FW: units activity min^−1^ g^−1^ of root’s fresh weight; within a column, means sharing the same letter are not significantly different at *p* ≤ 0.05 according to Tukey’s multiple range test.

**Table 4 jof-09-00037-t004:** Effect of filamentous fungi soil treatment on phenol, lipid peroxidation (MDA), and protein contents on tomato roots 7 days after *M. javanica* inoculation.

Treatment	MDA	Total Protein Content	Total Phenol Content
	µmol g^−1^ FW	mg g^−1^ FW	mg g^−1^ FW
Untreated Control	2.32 ± 0.99 ^c^	2.34 ± 0.18 ^c^	2.71 ± 0.14 ^bc^
RKN (N) only	12.44 ± 1.22 ^a^	3.36 ± 0.14 ^ab^	3.25 ± 0.23 ^a^
*T. longibrachiatum* + RKN	7.44 ± 1.01 ^b^	3.07 ± 0.08 ^abc^	2.01 ± 0.19 ^c^
*T. harzianum* + RKN	7.01 ± 0.92 ^b^	2.54 ± 0.26 ^abc^	2.52 ± 0.26 ^bc^
*T. asperellum* + RKN	6.97 ± 0.66 ^b^	3.12 ± 0.34 ^abc^	1.86 ± 0.43 ^c^
*M. chlamydosporia* + RKN	7.20 ± 0.46 ^b^	2.50 ± 0.43 ^bc^	2.32 ± 0.43 ^bc^
*Lecanicillium* spp. + RKN	10.76 ± 0.48 ^a^	2.93 ± 0.21 ^abc^	2.64 ± 0.32 ^abc^
CHM + RKN	7.09 ± 0.31 ^b^	2.62 ± 0.36 ^abc^	2.26 ± 0.43 ^bc^
BIO + RKN	7.34 ± 0.46 ^b^	3.49 ± 0.06 ^a^	2.84 ± 0.21 ^ab^

Values are mean of nine replicates (3 plants × 3 repetitions) ± SD; RKN: plants inoculated with *M. javanica* only; BIO: plants treated with abamectin; CHM: plants treated with ethoprophos; within a column, means sharing the same letter are not significantly different at *p* ≤ 0.05 according to Tukey’s multiple range test.

**Table 5 jof-09-00037-t005:** Root-knot nematode, *M. javanica,* development indices under various treatments, 60 days after inoculation.

Treatment	Gall Index	Galls Number/g of Root	Multiplication Rate (Pf/Pi)
RKN only	2.55 ± 0.54 ^a^	86.00 ± 2.92 ^a^	4.62 ± 1.67 ^a^
*T. longibrachiatum* + RKN	1.16 ± 0.26 ^b^	36.00 ± 1.56 ^c^	0.53 ± 0.87 ^d^
*T. harzianum* + RKN	1.83 ± 0.54 ^ab^	71.00 ± 2.10 ^b^	2.18 ± 0.65 ^c^
*T. asperellum* + RKN	2.00 ± 0.50 ^ab^	70.66 ± 2.32 ^b^	3.87 ± 0.42 ^b^
*M. chlamydosporia* + RKN	1.33 ± 0.51 ^b^	36.00 ± 3.49 ^c^	2.68 ± 0.81 ^c^
*Lecanicillium* spp. + RKN	2.33 ± 0.34 ^a^	67.83 ± 3.41 ^b^	3.33 ± 0.33 ^b^
CHM + RKN	1.16 ± 0.40 ^b^	39.00 ± 2.45 ^c^	0.54 ± 0.19 ^d^
BIO + RKN	1.5 ± 0.54 ^b^	41.50 ± 2.30 ^c^	0.64 ± 0.21 ^d^

Values are mean of twenty-one replicates (7 plants × 3 repetitions) ± SD; RKN: plants inoculated with *M. javanica*; BIO: plants treated with abamectin; CHM: plants treated with ethoprophos; within a column, means sharing the same letter are not significantly different at *p* ≤ 0.05 according to Tukey’s multiple range test.

## Data Availability

Not applicable.

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
