# Peer review of "Comparative Effectiveness of Filamentous Fungi in Biocontrol of Meloidogyne javanica and Activated Defense Mechanisms on Tomato"

_jof, 2022, doi:10.3390/jof9010037_

Round 1
Reviewer 1 Report
Investigations about the effectiveness of newly characterized BCAs is everytime desirable, as the case of the present manuscript. However, I have major concerns are about some technical and procedural issues, that have been listed below along with few minor points.
General comments: the main text must be checked for wrongly written species names (capiral letters etc.), syntax and punctuarion errors (i.e. ).
The most important issue os that it's not clear at all the rationale behind isolation of fungal strains from RKN-infested field in Turkey...Did Authors suggest that the best BCA could be obtained from infested soils? But how can it be explained? If an area results effectively infested, it's highly probable that resident microrganisms are definitely not so efficient as BCAs. With this regards, a second conceptual issue rises: even if Authors could have searched for good BCAs amongst the resident population of aforementioned soil, they reported that the 5 isolated strains have been used in the study, but they seemed to don't represent the best performing isolates - in terms of efficacy - resulting from a screening on a wider panel of taxa recovered. This is, in my opinion, a crucial point that must be seriously addressed, as the sole validation of a few casually isolated strains, when not corroborated by the comparison with commercially available or already validated strains and the providing of new findings or additional cues on BCA's mechanisms, make the experimental design of the manuscript almost weak.
Materials and methods:
The location of area mined for fungi isolation must be provided.
Line 189: "One gram of root material of each plant sampled 7 days after the RKN inoculations was ground " I wonder why root material has not been accuraley washed before protein extraction. In fact, in absence of a deep removal of possible fungal interacting micelium with plants, how could be excluded that a fraction of antioxidant enzymes activity can belong to fungal-derived proteins?
Line 192 reports " The supernatant was stored at -20ºC and used to determine the total soluble protein and enzymatic activity.". It's quite weird that for enzyme assays, that require optimal native conditions for the protein extracts to test, samples have been stored at -20°C instead of -80°C, as it must be... -20°C temperature in fact is well known to allow the formation of ice crystals that unfold and disrupt protein tertiary and quaternary polypeptide structures. If this really happened, a re-assessment of enzymatic activities should be performed in order to validate previously obtained results.
Line 216: "2.5.2. Total Soluble Protein Content" I' worried that the protocol used to determine total soluble proteins -as it has been performed- could have led to a misevaluation: clearly Authors retained to use same extracts dedicated to enzyme activity, but exactly for this reason the extraction protocol applied is not suitable for protein quantitation. Centrifugation has been conducted for only 10 minutes, that's not enough for a full sedientation of organellar and membrane fractions (at least 30 min at 14,000 x g are required, someties a second centrifugation is also necessary with plant tissues), extraction buffer lacks of beta-mercaptoethanol and MOPS, that help protein solubilization, and so on. I wonder if Authors could provide a re-assessment of samples or, in alternative, a re-arrangment of results presentation (for example a clarification about the fact that obtained protein values are not attributable to the total soluble protein content but to the native protein content of extracts used for enzyme assays) is necessary, with the coherent adjustments of graphs and discussion.
Lines 301-307: "The electron micrographs confirmed that.....and presumably digestion of its cellular contents" since a TEM analysis showing the inner structure of parasitized larvae hasn't been provided, I think that Authors should avoid any inference about the type of predatory activity occurring by the fungus, but they should limit their consideration to the observation that an interaction between hypae and larvae has been apparently established. Same as for the sentence "causing immobility and destructive damage to the overall outer cuticle layer." from abstract: since no clear evidence of such cuticle damage have been highlighted by reported images, also this quote must be deleted/changed.
Line 305: "could by trapping" I guess Authors wrongly put "by" instead of "be"
Figure 1: panel letters are partial; scale bars are absent; "100 μm red arrows" shouldn't be indicated like this: scale bars provided by the SEM image analysis software are the only required and requested as dimensional reference. Additionally, red and blue arrows that are reported to indicate neatode and myceliu respectively, actually aren't pointing to anything in particular. Please provide a correct version of the figure and fix the caption accordingly.
Table 2: "Values are mean of fourteen (2x7) replicates" please use 14 as the number is major than nine.
Line 330: "Antioxidant Enzyme Activities" change in "Antioxidant Enzymes Activity"
Line 334: "amplified the antioxidant enzymes" change in "amplified the activity of antioxidant enzymes"
Table 3 title: "Enzymatic activities" change in "Activity of antioxidant enzymes ascorbate peroxidase,....". Caption: "Values are mean of fourteen (2x7) replicates" please use 14 as the number is major than nine.
Paragraph 3.6: use "phenols" as plural
Figure 2: plase provide histogram with SD bars
Line 385: "numthe ber of" please fix
Line 390: "re-isolation of the fungal treatments" a fungal treatment can't be "re-isolated": fungal strains can be isolated/re-isolated, or fungi persistence can be assessed. However, the whole sentence is synctatically wrong. Please fix
Table 5. "Root knot nematode, M. javanica indices under various treatments" please correct punctuation. Additionally, 36±1.56 value is missing of the two decimals
Figure 3: please provide istogram with SD bars and species name in italic
Discussion: Authors began with statement "The results presented here indicate that M. javanica negatively affected the growth of tomato plants", supporting it with few literature. However, since the effect of the pathogen on tomato plants has been widely described (as cited works also highlight), I found the starting sentence alost mischoosen, 'cause it looks like this cue has been acquired here for the first time. Please delete or amend with a concept more consistent with the manuscript actual findings.
Line 452: "Indeed, phenolics are diverse secondary metabolites (including flavonoids, tannins, hydroxycinnamate esters, and lignin) that are abundant in plant tissues [81]. They possess antioxidant properties [72] and can directly scavenge molecular species of active oxygen [84]. Moreover, Michalak [85] presented some evidence of the induction of phenolic metabolism in plants in response to multiple stresses." references are inappropriate for the importance of the assertions, as the plant tissue content of metabolites, the scavenging potential of such compounds and the link between oxidative balance and plant stress response have been extremely investigated, described and assessed over decades. Please change citations with more consistent ones.
Line 473: "Tissues injured by oxidative stress generally contain increased concentrations of carbonylated proteins, which serve widely as a marker of protein oxidation [96]." move the sentence above, at line 469, and modify enumeration of references accordingly.
Conclusion: "The results indicate that fungal control using consortia of BCAs" I feel like this statement is partially incorrect, as the work did not explored the effects of association between the different fungal species investigated. On the contrary, only single BCA treatments have been studied.
Author Response
*General comments: the main text must be checked for wrongly written species names (capital letters etc.), syntax and punctuation errors (i.e.).
Dear sir everything is adjusted.
The most important issue of that it's not clear at all the rationale behind isolation of fungal strains from RKN-infested field in Turkey...Did Authors suggest that the best BCA could be obtained from infested soils? But how can it be explained? If an area results effectively infested, it's highly probable that resident microrganisms are definitely not so efficient as BCAs.
We appreciate the remarks of the reviewer and we are pleased to explain that the isolation of biological control agents in this study is based on RKN-infested soils in Tunisia because we are exploring the naturally occurring microorganisms living with Meloidogyne sp., and interacting with them by different ways e.g. suppressing the nematode population and reducing their infestation. So, the RKN rhizosphere could be the most promising agents to be used in controlling nematodes (those soils are suppressive soils and are characterized by low RKN infestation levels).
With this regards, a second conceptual issue rises: even if Authors could have searched for good BCAs amongst the resident population of aforementioned soil, they reported that the 5 isolated strains have been used in the study, but they seemed to don't represent the best performing isolates - in terms of efficacy - resulting from a screening on a wider panel of taxa recovered. This is, in my opinion, a crucial point that must be seriously addressed, as the sole validation of a few casually isolated strains, when not corroborated by the comparison with commercially available or already validated strains and the providing of new findings or additional cues on BCA's mechanisms, make the experimental design of the manuscript almost weak.
For this concern, we want to notice, first, the 5 fungal species selected for this study were screened from preliminary tests carried out in the laboratory and testing numerous fungal isolates. The most efficient and potential to be as BCAs were selected for deeper studies as this paper shows. In addition, the comparison of the new BCAs and commercial nematicides was done in our study and we compare their effectiveness with biological and chemical validated nematicides in pot experiment.
Materials and methods:
The location of the area mined for fungi isolation must be provided.
Done.
Line 189: "One gram of root material of each plant sampled 7 days after the RKN inoculations was ground " I wonder why root material has not been accurately washed before protein extraction. In fact, in absence of a deep removal of possible fungal interacting mycelium with plants, how could be excluded that a fraction of antioxidant enzyme activity can belong to fungal-derived proteins?
-We agree with the reviewer that the root material was washed before protein extraction and that was done during this study. We mentioned that actually in the manuscript as below ‘Three replicates of plants per treatment were uprooted 7 days after the nematode inoculation and were washed with distilled water to carry out the biochemical analysis.’
Line 192 reports "The supernatant was stored at -20ºC and used to determine the total soluble protein and enzymatic activity." It's quite weird that for enzyme assays, that require optimal native conditions for the protein extracts to test, samples have been stored at -20°C instead of -80°C, as it must be... -20°C temperature in fact is well known to allow the formation of ice crystals that unfold and disrupt protein tertiary and quaternary polypeptide structures. If this really happened, a re-assessment of enzymatic activities should be performed in order to validate previously obtained results.
We agree with the reviewer and we review and correct the manuscript. The enzymatic extract was stored at -80°C
Line 216: "2.5.2. Total Soluble Protein Content" I’m worried that the protocol used to determine total soluble proteins -as it has been performed- could have led to a misevaluation: clearly Authors retained to use same extracts dedicated to enzyme activity, but exactly for this reason the extraction protocol applied is not suitable for protein quantitation. Centrifugation has been conducted for only 10 minutes, that's not enough for a full sedimentation of organellar and membrane fractions (at least 30 min at 14,000 x g are required, sometimes a second centrifugation is also necessary with plant tissues), extraction buffer lacks of beta-mercaptoethanol and MOPS, that help protein solubilization, and so on. I wonder if Authors could provide a re-assessment of samples or, in alternative, a re-arrangment of results presentation (for example a clarification about the fact that obtained protein values are not attributable to the total soluble protein content but to the native protein content of extracts used for enzyme assays) is necessary, with the coherent adjustments of graphs and discussion.
The root protein extraction was assessed by TCA/acetone method as described by Xu et al. (2008). We add the reference of (XU C, XU Y AND HUANG B. 2008. Protein extraction for two-dimensional gel electrophoresis of proteomic profiling in turfgrass. CropSci 48: 1608-1614)
Lines 301-307: "The electron micrographs confirmed that.....and presumably digestion of its cellular contents" since a TEM analysis showing the inner structure of parasitized larvae hasn't been provided, I think that Authors should avoid any inference about the type of predatory activity occurring by the fungus, but they should limit their consideration to the observation that an interaction between hypae and larvae has been apparently established. Same as for the sentence "causing immobility and destructive damage to the overall outer cuticle layer." from abstract: since no clear evidence of such cuticle damage have been highlighted by reported images, also this quote must be deleted/changed.
We added this phrase to explain ’The fungal hyphae and conidia adhered to the juveniles’cuticle layer. They formed trapping-rings around emerging RKN juveniles and they cause their immobility.
Line 305: "could by trapping" I guess Authors wrongly put "by" instead of "be"
We review it and replaced ‘by’ by ‘be’
Figure 1: panel letters are partial; scale bars are absent; "100 μm red arrows" shouldn't be indicated like this: scale bars provided by the SEM image analysis software are the only required and requested as dimensional reference. Additionally, red and blue arrows that are reported to indicate neatode and myceliu respectively, actually aren't pointing to anything in particular. Please provide a correct version of the figure and fix the caption accordingly.
We adjusted SEM micrographs as recommended (scale bars completed, complete panel letters and adjust arrows to indicate nematode juveniles and fungal phyphae and conidia)
Table 2: "Values are mean of fourteen (2x7) replicates" please use 14 as the number is more major than nine.
We reviewed the number of plants is twenty-one (7 tomato plants replicated 3 times: we added (7*3 to explain)
Line 330: "Antioxidant Enzyme Activities" change in "Antioxidant Enzymes Activity"
Changed
Line 334: "amplified the antioxidant enzymes" change in "amplified the activity of antioxidant enzymes"
Changed
Table 3 title: "Enzymatic activities" change in "Activity of antioxidant enzymes ascorbate peroxidase,....". Caption: "Values are mean of fourteen (2x7) replicates" please use 14 as the number is major than nine.
Done
Paragraph 3.6: use "phenols" as plural
They are plural
Figure 2: please provide histogram with SD bars
Done
Line 385: "numtheber of" please fix
Fixed
Line 390: "re-isolation of the fungal treatments" a fungal treatment can't be "re-isolated": fungal strains can be isolated/re-isolated, or fungi persistence can be assessed. However, the whole sentence is synctatically wrong. Please fix
Fixed
Table 5. "Root knot nematode, M. javanica indices under various treatments" please correct punctuation. Additionally, 36±1.56 value is missing of the two decimals
Done
Figure 3: please provide istogram with SD bars and species name in italic
Done
Discussion: Authors began with statement "The results presented here indicate that M. javanica negatively affected the growth of tomato plants", supporting it with few literature. However, since the effect of the pathogen on tomato plants has been widely described (as cited works also highlight), I found the starting sentence almost mischoosen, 'cause it looks like this cue has been acquired here for the first time. Please delete or amend with a concept more consistent with the manuscript actual findings.
Done
Line 452: "Indeed, phenolics are diverse secondary metabolites (including flavonoids, tannins, hydroxycinnamate esters, and lignin) that are abundant in plant tissues [81]. They possess antioxidant properties [72] and can directly scavenge molecular species of active oxygen [84]. Moreover, Michalak [85] presented some evidence of the induction of phenolic metabolism in plants in response to multiple stresses." references are inappropriate for the importance of the assertions, as the plant tissue content of metabolites, the scavenging potential of such compounds and the link between oxidative balance and plant stress response have been extremely investigated, described and assessed over decades. Please change citations with more consistent ones.
Done
Line 473: "Tissues injured by oxidative stress generally contain increased concentrations of carbonylated proteins, which serve widely as a marker of protein oxidation [96]." move the sentence above, at line 469, and modify enumeration of references accordingly.
Done
Conclusion: "The results indicate that fungal control using consortia of BCAs" I feel like this statement is partially incorrect, as the work did not explored the effects of association between the different fungal species investigated. On the contrary, only single BCA treatments have been studied.
We agree with the reviewer, and we corrected it that fungal BCAs are used individually and not in consortia
Reviewer 2 Report
The manuscript reports the comparative effectiveness of five filamentous fungi in biocontrol of M. javanica and activated defense mechanisms on tomato host.The work is of interest, and the test is well designed and conducted. There are several queries that I would like to pose to the authors.
1. In line176, it is suggested to give a detailed description on the preparation of the spore suspension.
2. In table 3, the results of the difference of significant analysis seem somewhat strange. For example, about the activities of CAT, there is a significant difference between the treatments of T. longibrachiatum+N (22.12±0.61) and T. asperellum + N (19.4±0.34), but T. longibrachiatum+N showed no significant difference with T. harzianum + N(18.64±0.82), since the value of 19.4 more close to 22.12. The similar results also found in Table 2. Please explain why.
3. In figure 3, lack of the letter ‘d’ in marking the significant difference. Moreover, this figure doesn’t like an result of the M. javanica hatching, it different to the illustration in line 393.
Author Response
Reviewer 2
The manuscript reports the comparative effectiveness of five filamentous fungi in biocontrol of M. javanica and activated defense mechanisms on tomato host. The work is of interest, and the test is well designed and conducted. There are several queries that I would like to pose to the authors.
- In line176, it is suggested to give a detailed description on the preparation of the spore suspension.
We added this paragraph "For BCAs fungi, the species were grown on PDA at 25±2°C for five days until sporulation, and then, an Erlenmeyer flask containing 50 ml of potato dextrose broth was inoculated with four pieces individually. Spore production was induced in an orbital shaker, and the spores were recovered from culture by filtration. A hemacytometer served to adjust the concentrations of the spores to 4 x106 spores/ml."
- In table 3, the results of the difference of significant analysis seem somewhat strange. For example, about the activities of CAT, there is a significant difference between the treatments of T. longibrachiatum+N (22.12±0.61) and T. asperellum + N (19.4±0.34), but T. longibrachiatum+N showed no significant difference with T. harzianum + N(18.64±0.82), since the value of 19.4 more close to 22.12. The similar results also found in Table 2. Please explain why.
Done.
- In figure 3, lack of the letter ‘d’ in marking the significant difference. Moreover, this figure doesn’t like an result of the M. javanica hatching, it different to the illustration in line 393.
Done.
Reviewer 3 Report
Comments:
The paper entitled “Comparative effectiveness of filamentous fungi in biocontrol of Meloidogyne javanica and activated defense mechanisms on tomato” has been critically checked. The paper contains good information, but it is not presented well. The Scientific names should always be in italic. The English of the manuscript is very poor, and also there are lot of grammatical errors. Also, the paper is not set according to the Journal guidelines and in the references, the name of some journals is abbreviated and some written in full form, it should be according to Journal guidelines. Some references are written with capitalise each word format, which is not the right way. Also, I wonder why the submitted manuscript has text highlight colour in some paragraphs. The species name is always written small which is not the case in the present manuscript. The paper needs a lot of improvement and it cannot be accepted in the Journal of Fungi in the present.
Some of the major errors noticed in the manuscript are given below and most of the other comments and changes are given with attached edited PDF.
Line 99-105: How do authors say that the isolated RKN is M. javanica without any morphological or molecular studies? They have not mentioned anything about the characterization of RKN.
Line 128-139; 273-278: Authors of the present study have used ITS rRNA for molecular characterisation of fungal taxa, but I do not understand why authors presented data of 28rRNA in result section. This thing authors should have checked carefully.
Line 106; 262-278: Authors have used phenotypic identification of fungal taxa in the heading, but they have written how they did phenotypic identification and also, they have not mentioned anything about phenotypic identification of fungal taxa in the result section.
Line 351: Authors have not mentioned about V. lecani in the materials and methods section then how they write that the MDA content in the tomato roots decreased significantly after the application of all of the treatments except with V. lecani in the line 351. This is confusing.
Line 385: By reading this sentence “The development of the nematodes was measured as the gall index, numthe ber of galls per g of root, and the multiplication rate”, it looks that all the authors have not read this manuscript before submission.
Figures 2 and 3 also need to be improved. Authors have used of six replicates in their assays, then why there are not significant bars used in the graphs.

Author Response
Reviewer 3
The paper entitled “Comparative effectiveness of filamentous fungi in biocontrol of Meloidogyne javanica and activated defense mechanisms on tomato” has been critically checked. The paper contains good information, but it is not presented well.
The Scientific names should always be in italic.
Done
Also, the paper is not set according to the Journal guidelines and in the references, the name of some journals is abbreviated and some written in full form, it should be according to Journal guidelines. Some references are written with capitalise each word format, which is not the right way.
Revised based on the journal guidelines
Also, I wonder why the submitted manuscript has text highlight color in some paragraphs.
Removed
The species name is always written small which is not the case in the present manuscript.
Corrected
Line 99-105: How do authors say that the isolated RKN is M. javanica without any morphological or molecular studies? They have not mentioned anything about the characterization of RKN.
Added to the text
Line 106; 262-278: Authors have used phenotypic identification of fungal taxa in the heading, but they have written how they did phenotypic identification and also, they have not mentioned anything about phenotypic identification of fungal taxa in the result section.
Dear respected reviewer, phenotypic identification of fungal taxa all the time used the relevant identification keys which already mentioned in the materials and methods section. All the five taxa recovered are identified phenotypically and confirmed by molecular means and revised in the result section.
Line 351: Authors have not mentioned about V. lecani in the materials and methods section then how they write that the MDA content in the tomato roots decreased significantly after the application of all of the treatments except with V. lecani in the line 351. This is confusing.
We replaced "V. lecani" with "Lecanicillium spp."
Line 385: By reading this sentence “The development of the nematodes was measured as the gall index, number of galls per g of root, and the multiplication rate”, it looks that all the authors have not read this manuscript before submission.
Revised
Figures 2 and 3 also need to be improved. Authors have used of six replicates in their assays, then why there are not significant bars used in the graphs.
Done.
Reviewer 4 Report
Please revise the manuscript as per the attached file
regards

Author Response
Reviewer 4
Rose Bengal is a bactericide or fungicide
Rose Bengal is a bacteriostatic to suppress bacterial growth along with bactericidal chloramphenicol.
Why i did not see any phylogenetic tree constructed
Dear respected reviewer, the phylogenetic tree will not add anything new to the work as the recovered taxa already deposited in GenBank.
- why this kind of microscope has low magnification power. Do you see something like the egg hatching????
- it is the same name of dissecting microscope or another??
- it is correct?? ‘infected eggs (colonized with fungi hyphae)
We used a Leica M80 with led up and down object illumination system, and 60x magnification + 8 zooms. These characteristics permit to visualization of the juveniles, the eggs, the hatching as well as the fungi hyphae colonizing eggs and larva. we use it for counting because it gives a large spectrum of visualization (even the entirety of the Petri dish) and ease in following the movement of the object to be observed.
P6- M and M : did not match with the results
Thank you for this comment and in fact there was a writing mistake in the figure which was corrected and we replaced egg number by the juvenile’s number as we counted juveniles, not eggs.
Delete this image A it is the same with B
It’s the same picture but at different magnification: Picture A gives a global vision of the infection by the fungi hyphae an B presents more details
The arrows very confused i didn't see what you mean by your arrows???
The arrow as explained in the bottom of the pictures differentiates Nematodes and fungi. Red ones represents the nematode’s juvenile and the green ones present the fungi
How many replicates do you have? Please confirm it as you mentioned in the M and M section. it is so confusing
Replicates :
In vitro test:
- effect of fungi on RKN: 6 replicates x 3 repetitions =18
- Eggs fecundity after 60 days harvesting: 2ggs / trt x 3 replicates x2 repetitions =12 eggs/trt
Pot experiment:
- 7 days (enzyme’s activity): 3 pots x 3 repetitions =9replicates
- 60 days: 7 pots x 3 repetitions=21 replicates
add the name of pesticide and repeat that in all the manuscript
Done
could you write the unit as the same in all the manuscript. plz unit the unit
Done
what this is ??? is it 7 days or 60 days the methods told 7 days
Enzyme’s study= 7 days
what this is ??????? plz delete with the figure 2
Deleted.
Round 2
Reviewer 1 Report
Authors convincingly answer to my concerns about the rationale of the study, but I would like that these clarifications -as they have been reported to me: " the isolation of biological control agents in this study is based on ...... So, the RKN rhizosphere could be the most promising agents to be used in controlling nematodes (those soils are suppressive soils and are characterized by low RKN infestation levels)." and "the 5 fungal species selected for this study were screened ...... nematicides in pot experimen"- will be also added to the manuscript for a better comprehension.
Author Response
Dear respected reviewer all your point are answered in the attached file

Reviewer 3 Report
Comments:
The revised manuscript lacks line numbers, which makes it difficult for review process although previous version was uploaded with line numbers. The authors have not revised the manuscript well and the comments and suggestions which I gave in the previous attached edited pdf looks as if authors have not been looked on them. The typing errors are still there which authors should have at least corrected in the revised version. Some of the major comments which authors have not addressed are listed below. Also, I request authors to look at the edited pdf version which I attached previously and now attaching again for your reference.
1. Previously, I requested authors to clear about how they say that the isolated RKN is M. javanica without any morphological or molecular studies. In the revised manuscript, the authors have mentioned that Identification of the RKN, Meloidogyne javanica (Treub) Chitwood was based on morphological characteristics of Perineal pattern of the adult females according to Taylor and Sasser in the materials and methods section, but I wonder why authors have not mentioned anything about Perineal pattern of the adult females in the result or discussion part. Further, I as being a taxonomist, it seems for me also difficult to identify Meloidogyne up to species level only on basis of perineal pattern since this genus contains a lot of species. Currently, there are molecular tools available to further authenticate the identification of species which authors should have used.
2. The other query that the authors have not addressed that I also commented previously is that in the methodology, authors have used primers for sanger sequencing of ITS rRNA (ITS1-5.8S-ITS2), but I wonder why authors discuss about analysis of 28S rRNA sequences in the result section. When I also nucleotide blasted the submitted sequences of the present study, the analysis results were ok, but only mistake is that they are the results ITS rRNA sequences and not 28S rRNA sequences which authors should have edited.
3. In section 3.1, Metacordyceps chlamydosporia (H.C. Evans) G.H. Sung, J.M. Sung, Hywel-Jones & Spatafora, ... the authors should write the authority names correctly as we always use last names only within the text when cited and full names in the references. I think authors have not gone through my previously uploaded edited pdf version.
4. The other minor comments and suggestions are in the attached edited Pdf.

Author Response

(The authors gave the same response as above.)

Reviewer 4 Report
Since your paper was submitted to the Journal of Fungi, you must use other genes to confirm your isolates' identities, just like the rpb2 or tef-1 genes.
Even your enzyme results failed to persuade me. I wonder why the authors did not do additional experiments on the molecular level since their paper title concluded that the words "activated defense mechanisms," which need more work to establish, are not established by enzymatic activity studies alone, so my suggestion is that you must do RT-qPCR for all samples along the phenylpropanoid pathway genes just like CHS, HQT, CHI, PAL, and AN1 OR AN2 to confirm your results.
Author Response
Dear respected reviewer our reply on your request on the attached file
